# Multi-supervised bidirectional fusion network for road-surface condition recognition

Hongbin Zhang[1], Zhijie Li[1], Wengang Wang[1], Lang Hu[1], Jiayue Xu[2], Meng Yuan[1], Zelin Wang[3], Yafeng Ren[4] and Yiyuan Ye[5]

[1] School of Software, East China JiaoTong University, Nanchang, China
[2] School of Business School, Changzhou University, Changzhou, China
[3] School of Information Science and Technology, Nantong University, Nantong, China
[4] School of Interpreting and Translation Studies, Guangdong University of Foreign Studies, Guangzhou, China
[5] School of Information Engineering, East China Jiaotong University, Nanchang, China



Corresponding author
Hongbin Zhang, 1332@ecjtu.edu.cn

## ABSTRACT

Rapid developments in automatic driving technology have given rise to new experiences for passengers. Safety is a main priority in automatic driving. A strong familiarity with road-surface conditions during the day and night is essential to ensuring driving safety. Existing models used for recognizing road-surface conditions lack the required robustness and generalization abilities. Most studies only validated the performance of these models on daylight images. To address this problem, we propose a novel multi-supervised bidirectional fusion network (MBFN) model to detect weather-induced road-surface conditions on the path of automatic vehicles at both daytime and nighttime. We employed ConvNeXt to extract the basic features, which were further processed using a new bidirectional fusion module to create a fused feature. Then, the basic and fused features were concatenated to generate a refined feature with greater discriminative and generalization abilities. Finally, we designed a multi-supervised loss function to train the MBFN model based on the extracted features. Experiments were conducted using two public datasets. The results clearly demonstrated that the MBFN model could classify diverse road-surface conditions, such as dry, wet, and snowy conditions, with a satisfactory accuracy and outperform state-of-the-art baseline models. Notably, the proposed model has multiple variants that could also achieve competitive performances under different road conditions. The code for the MBFN model is shared at https://zenodo.org/badge/latestdoi/607014079.

## INTRODUCTION

Globally, traffic accidents cause economic losses equivalent to 600 billion USD annually. Automatic driving technology can improve driving safety and the efficiencies of transportation systems. Advanced obstacle detection systems are required to ensure the safety and comfort of the passengers in automatic vehicles. One of the key tasks of an

obstacle detection system is to accurately detect different road-surface conditions, such as dry, wet, and snow-covered road-surface conditions, on the path of an automatic vehicle. As we know, the risk of traffic accidents is significantly related to weather conditions (*Zhang et al., 2021a*; *Bellone et al., 2021*). According to the European Road Safety Observatory, 29% of the fatal accidents in 2016 occurred under non-dry road-surface conditions, including rain, fog, and snow (*European Road Safety Observatory, 2018*), which reduced the road grip and increased the braking distances of vehicles. Therefore, the accurate and efficient classification of road-surface conditions is crucial for ensuring safety in automatic driving. Advantageously, a self-driving vehicle can refer to the recognition results and brake in advance whenever necessary, improving overall safety.

Road-surface condition recognition has been investigated since the 1990s (*Fukui et al., 1997*; *Chen, 1991*; *Holzwarth & Eichhorn, 1993*). A recent technique for classifying road-surface conditions exploits the variations in the intensity of scattered near-infrared (NIR) light from road-surfaces. *Ruiz-Llata et al. (2017)* and *Casselgren, Sjödahl & LeBlanc (2012)* investigated the feasibility of NIR systems for classifying road-surface conditions through a system that distinguished road-surface conditions using three wavelengths originating from a NIR band, whereas this system was cost-prohibitive, limiting its widespread applicability. More researchers have been using machine learning technology, owing to their increasing development. *Zhao, Wu & Chen (2017)* used support vector machines (SVMs) to classify road-surface conditions (*Zhao, Wu & Chen, 2017*; *Omer & Fu, 2010*), achieving an accuracy of approximately 90%, which was considered unsatisfactory for ensuring safety in autonomous driving. *Marianingsih & Utaminingrum (2018)* pointed out that SVMs outperformed naive Bayes classifiers in road condition classification. *Fauzi, Utaminingrum & Ramdani (2020)* proposed a method that combined a gray level co-occurrence matrix and local binary pattern features to improve the robustness of the recognition model. The development of deep learning methods, such as convolutional neural networks (CNN), from 2012 has significantly advanced research on computer vision (*Krizhevsky, Sutskever & Hinton, 2017*). Deep learning is an emerging method that plays an important role in road-surface condition recognition. Some deep-learning methods focus on recognizing wet and dry road conditions. Overall, significant progress has been achieved in the field of autonomous driving. Regarding a single deep-learning network, some important contextual information may be lost during the feature-learning procedure, resulting in a poor performance. Moreover, conditions with low brightness or contrast can result in images of poor quality that limit the generalization ability of current methods in obtaining sufficient discriminative information, resulting in great challenges in recognizing night road-surface conditions.

To address these issues, we propose a new model called multi-supervised bidirectional fusion network (MBFN). First, the state-of-the-art ConvNeXt (*Liu et al., 2022*) model was employed as the backbone for basic feature generation. Using this model, discriminative and robust features were obtained in the subsequent stages. Second, to address the loss of contextual information, a novel bidirectional fusion module was designed to integrate the contextual information and create a fused feature. Then, the fused and basic features were combined to generate a refined feature. Finally, multi-supervised signals were employed to

train the classification model. Conceptually and empirically, the main contributions of this study are as follows.

1) The MBFN model has been proposed for the effective and robust recognition of road-surface conditions.
2) A bidirectional fusion module that can fuse diverse features from different stages and fully utilize their complementary information has been designed.
3) Extensive experiments were conducted using two benchmark datasets. The corresponding results demonstrated that the MBFN model outperformed state-of-the-art baseline models. The MBFN model could clearly recognize both daytime and nighttime road-surface conditions. Moreover, the generalization ability of the proposed model was validated using a coarse-grained material image dataset.
4) Notably, the proposed model had multiple variants, and the performance of these variants was also satisfactory, demonstrating the effectiveness of the model for road-surface condition recognition.

The rest of the article is organized as follows. Section 2 introduces related literature and our research motivations. Section 3 describes the proposed MBFN model. Then, sections 4 and 5 discuss the experimental results for the three benchmark datasets. Finally, section 6 presents the conclusions and the scope for future research.

## RELATED WORK

### Road-surface condition recognition

In recent years, significant advancements have been made in automatic driving technology, and more people are relying on the safety and practical applicability of this technology. Improvements in advanced machine learning technologies facilitate the easy analysis of road-surface conditions (*Singh & Arat, 1906*). *Kim, Lee & Yoon (2021)* used weather station data to predict rainy road conditions. Their method required considerable human resources. Deviations in human measurements often result in unexpected uncertainties. *Smolyakov & Burnaev (2020)* used the model of the environment and temperature of roads to predict icy-road conditions. This model combined a physical model for predicting road conditions based on site measurements with a machine learning model to detect incorrect data. Other researchers have focused on data collected from cameras because visual information is the most intuitive for accurately distinguishing road-surface conditions. *Amthor, Hartmann & Denzler (2015)* proposed a texture-based model for detecting wet road conditions. *Zhao, Wu & Chen (2017)* and *Omer & Fu (2010)* used a SVM to classify road-surface conditions, obtaining an accuracy of approximately 90%, which was still considered unsatisfactory considering the safety required in self-driving.

Lately, many deep learning-based methods have been proposed for road-surface condition recognition. *Roychowdhury et al. (2018)* achieved an accuracy of 97% using SqueezeNet. *Fink et al. (2020)* went one step further by using SqueezeNet to reduce the

computational complexity without significantly affecting the accuracy. *Svensson (2020)* and *Balcerek et al. (2020)* utilized DenseNet121 and AlexNet to classify road-surface conditions and estimate the corresponding road friction. *Dewangan & Sahu (2021)* proposed a CNN model called RCNet to classify road-surface conditions, achieving an accuracy of 99%. To solve the data imbalance problem, *Choi, Heo & Ahn (2021)* proposed a generative adversarial network-based model to generate more images of wet and snow-covered road-surface conditions, reducing data imbalance and achieving a satisfactory classification performance.

The above studies mainly focused on the recognition of road-surface conditions during the daytime, neglecting conditions at nighttime. Slippery road conditions are more challenging to detect at night because of the limited lighting conditions. There is a sharp reduction in the contrast of road-surface images at night, which can potentially increase the risk of accidents. This clearly shows the pressing need of obstacle systems for detecting road-surface conditions at nighttime. *Shibata et al. (2014)* and *Horita et al. (2012)* investigated this challenging research direction using camera images at nighttime. They used the Mahalanobis distance to distinguish different road-surface conditions, achieving a recognition accuracy of only 70–80%. *Kawai et al. (2012)* extracted the brightness, color information, and texture features of camera images at nighttime and used the nearest neighbor algorithm to recognize road-surface conditions, obtaining an accuracy of approximately 90%, which was still far from practical requirements. Another disadvantage in their studies was that the collected datasets were limited to a given road, and the test data were obtained on the same road, limiting the robustness of the model in other scenarios.

Recognizing road-surface conditions at nighttime is challenging, which can be attributed to images captured at night being highly dependent on road lighting conditions. The main sources of illumination at night are ambient light sources, such as vehicle headlights and streetlights. The differences in the positions of these light sources relative to the camera used for capturing them result in the images having different characteristics when illuminated by two light sources. Hence, the extracted features are not discriminative and robust under road-surface conditions. Meanwhile, because of continuous down-sampling, some important contextual information gets lost. Overall, a more delicate feature-processing method is required to alleviate these problems.

## Research motivations

Our research has three motivations. First, an effective deep learning framework was required to extract basic features with a certain reliability and high quality and thereby create a solid foundation for subsequent feature refinement; ConvNeXt was selected for this purpose. Second, semantically complementary and robust features should be learned at different scales to mitigate the effects of light and continuous down-sampling. Therefore, basic features were inputted into the bidirectional fusion module to create a fused feature, which was then combined with the basic features to generate refined features. Finally, we used the basic and refined features to coordinate the bidirectional fusion module and ConvNeXt, ensuring the improved model faster convergence.

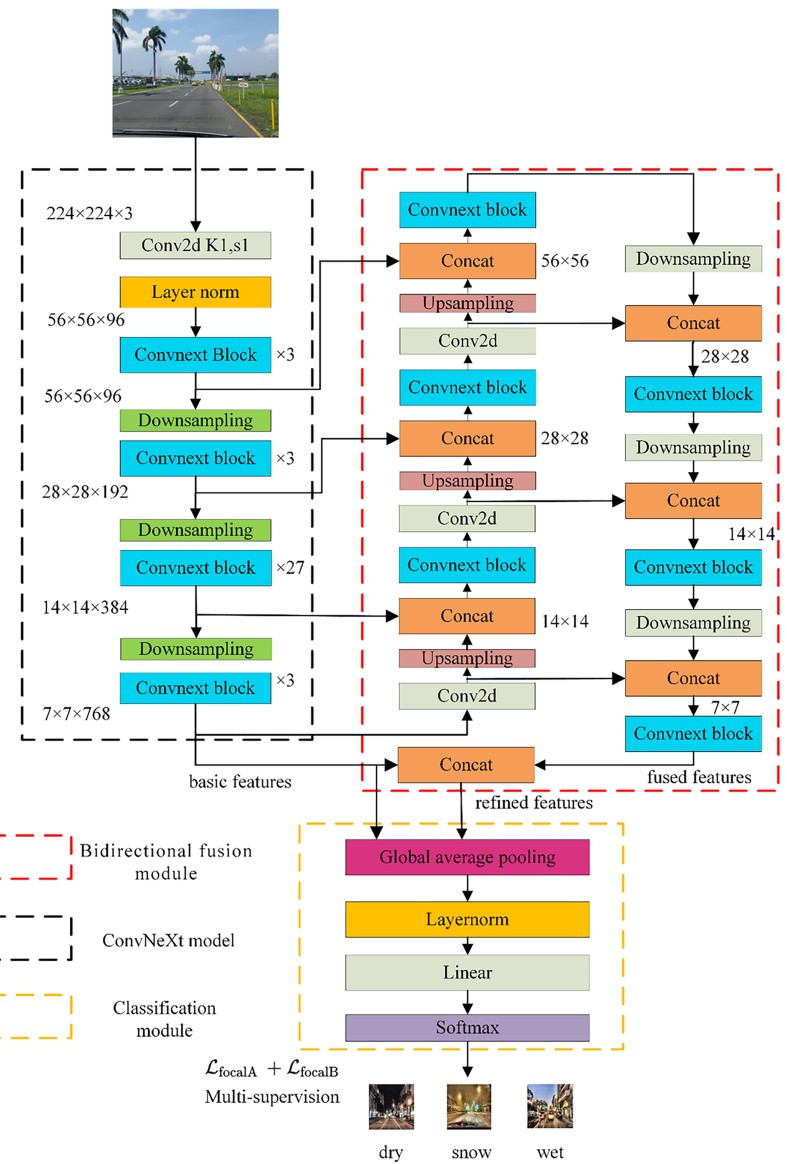

**Figure 1** **The model structure of MBFN.** Image source: YouTube (https://www.youtube.com/playlist?list=PLZi94yHOS-GROyCk6zvEnNb0LbCyCpc1O).

## PROPOSED MBFN MODEL

The proposed method is shown in Fig. 1. The core concept of the MBFN model is effectively learning semantically complementary and robust features at different scales to alleviate the problem of semantic information loss during continuous down-sampling. These new features can be applied in both daylight and nighttime scenarios. The MBFN model takes a 224 × 224 road-surface image as the input and obtains basic features through the backbone network, namely ConvNeXt (*Liu et al., 2022*). Then, a bidirectional fusion module, in which an up-sampling process is first implemented, is employed. The features output at different stages are spliced and fused with the features generated by ConvNeXt, enabling richer semantic information to be obtained. Second, motivated by PANet (*Liu*

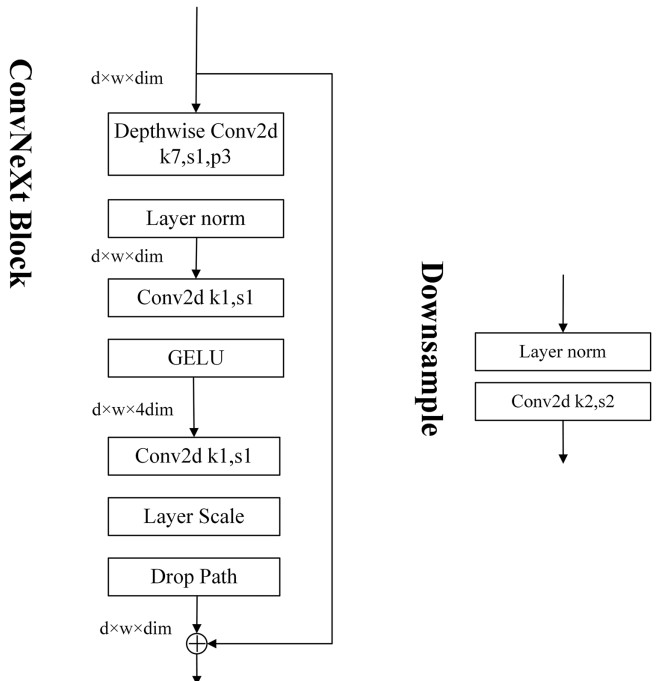

**Figure 2  The model structure of ConvNeXt block.**

*et al., 2018*), top-down down-sampling is performed, and features from different stages in the up-sampling and down-sampling operations are fused, resulting in fused features with sufficient semantic information. Behind the bidirectional fusion module, the basic and fused features are fused to obtain the refined features. Finally, a classification module consisting of a global average pooling layer, a fully connected layer, and a softmax classifier is designed, and the multi-supervised loss embedded in the classification module is utilized to accomplish road-surface condition classification.

## ConvNeXt

We chose the popular ConvNeXt (*Liu et al., 2022*) network, which was released in 2022, as the backbone network for basic feature extraction. The model provided satisfactory results despite not having a complicated structure and provided competitive results compared to those of Transformer. The network structure of ConvNeXt is shown in Fig. 1.

There are five versions of ConvNeXt, namely ConvNeXt-T, ConvNeXt-S, ConvNeXt-B, ConvNeXt-L, and ConvNeXt-XL. In this study, we used ConvNeXt-S, which has a ConvNeXt Block with modules stacked in the following numbers: 3, 3, 27, and 3. The structure of the ConvNeXt Block is shown in Fig. 2.

## Bidirectional fusion module

As we know, feature pyramid network (*Lin et al., 2016*) (FPN) constructs a multi-layer feature pyramid, that is, it extracts feature information of different scales from different feature layers. Hence, FPN outputs feature maps of multiple scales and fuses these feature maps through top-down and bottom-up paths to generate a feature pyramid with plentiful

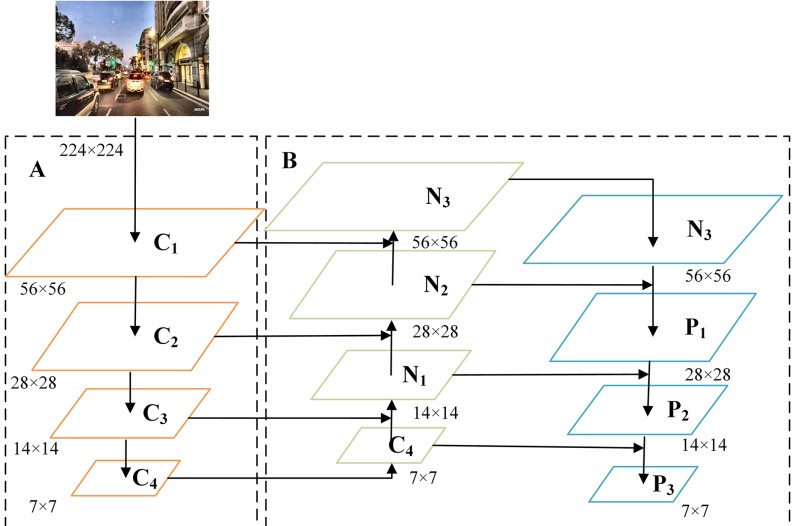

**Figure 3 The overall simplified model structure of MBFN.** Image source: YouTube (https://www.youtube.com/playlist?list=PLZi94yHOS-GROyCk6zvEnNb0LbCyCpc1O).

multi-scale information. Notably, it passes down the high-level features and supplement the low-level semantics so that high-resolution but more discriminative semantics information can be obtained. Since continuous down-sampling operations may lose some significant discriminative information, we designed a structure like FPN (*Zhang et al., 2022*) to address this problem in our MBFN model. We performed an up-sampling operation on the final layer of the ConvNeXt model. Thus, we get the features of the same scale as the down-sampling stage of the ConvNeXt network. The two features come from different stages are concatenated in turn to create a new feature with sufficient semantic information.

A simplified structure of the MBFN is shown in Fig. 3. Part A is the ConvNeXt network, and part B is the proposed bidirectional fusion module. The left part of the bidirectional fusion module performs up-sampling restoration, resembling the working of FPN. The essential difference is that during up-sampling, a simple splicing operation is performed on the features at the same scale as in the down-sampling process of ConvNeXt, enabling the generated features to effectively retain crucial discriminative information. The right side of the bidirectional fusion module is a bottom-up complement, which performs down-sampling like a pyramid structure does. The features obtained by down-sampling are fused with the features at the same scale obtained in middle up-sampling. Finally, the fused features are obtained, following which they are combined with the basic features to obtain the refined features. The splicing process is illustrated in Fig. 4. The fusion operation is performed using a 3 × 3 convolution, which improves the representation ability of the generated features.

## Multi-supervised loss
We replaced the traditional cross-entropy loss with the focal loss (*Lin et al., 2017*), which can solve training problems caused by sample imbalance. The cross-entropy loss adds a

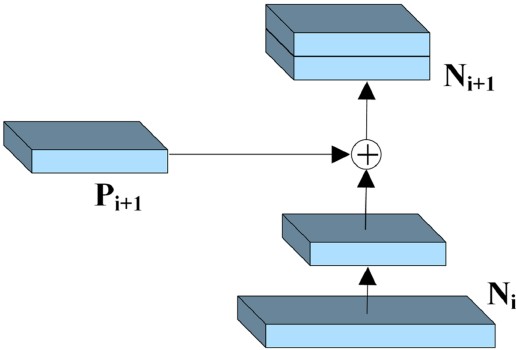

**Figure 4  The model fusion process.**     

weight factor to the loss function from the perspective of sample distribution. The focal loss originates from the difficulties in sample classification and shifts the focus of the loss function to the difficult samples. The focal loss is expressed as follows:

$$\mathcal{L}_{fl} = \begin{cases} -(1-\hat{p})^{\gamma} log(\hat{p}) & if\ y = 1 \\ -\hat{p}^{\gamma} log(1-\hat{p}) & if\ y = 0 \end{cases} \tag{1}$$

where the order $p_t$ is:

$$p_t = \begin{cases} \hat{p} & if\ y = 1 \\ 1-\hat{p} & otherwise \end{cases} \tag{2}$$

Then, the focal loss expression can be unified as:

$$\mathcal{L}_{fl} = -(1-p_t)^{\gamma} \log(p_t) \tag{3}$$

where $\gamma$ is an adjustable factor; and $p_t$ reflects the closeness to the ground truth (the larger the $p_t$ value, the closer it is to category $y$) and the difficulty of the classification, with $p_t$ being directly proportional to the classification confidence. Therefore, the focal loss assigns higher weighs to more difficult samples, improving the final accuracy.

To weigh the effects of ConvNeXt and the bidirectional fusion modules, we used two loss functions, focalA and focalB, to supervise the two modules. The proposed model employed two focal losses to obtain the multi-supervised loss as follows:

$$\mathcal{L} = \mathcal{L}_{focalA}(A, A') + \mathcal{L}_{focalB}(B, B') \tag{4}$$

where $\mathcal{L}_{focalA}(A, A')$ is the focal loss between the predicted label $A$ and ground-truth label $A'$, and $\mathcal{L}_{focalB}(B, B')$ is the focal loss between the predicted label $B$ and ground-truth label $B'$.

## DATASET PREPARATION

Since no specific database or dataset focuses on road-surface conditions at night, an individual dataset for these conditions is required. Data from several YouTube videos were collected to form the dataset in *Zhang et al. (2022)*. Using this dataset, the proposed model can be applied to diverse scenarios. Each image was extracted from the video frames at an interval of at least 1 s between the frames to ensure the variability within the dataset, and

**Table 1 Data set display.**

| Dataset | Training | | | Testing | | |
|---|---|---|---|---|---|---|
| | Dry | Wet | Snow | Dry | Wet | Snow |
| YouTube-w-ALI | 3,219 | 3,464 | 3,510 | 3,722 | 2,837 | 4,222 |
| YouTube-w/o-ALI | 3,722 | 3,830 | 3,601 | 6,081 | 2,809 | 4,183 |

**Table 2 MattrSet dataset.**

| Category | Bag_pu | Bag_canvas | Bag_nylon | Bag_polyester | Shoe_pu | Shoe_canvas |
|---|---|---|---|---|---|---|
| Number | 1,982 | 1,948 | 3,510 | 1,715 | 1,757 | 1,855 |

the extracted images were labeled manually. The corresponding videos are available at https://doi.org/10.6084/m9.figshare.22775078. It is common knowledge that videos taken with ambient light illumination (ALI) differ from those taken without it; videos taken with ALI are often taken from urban areas, while those without it are usually taken from the countryside or highways. Therefore, the dataset was collected separately under two illumination conditions, resulting in two datasets being created: YouTube-w-ALI and YouTube-w/o-ALI. The resized and preprocessed datasets are available at https://doi.org/10.6084/m9.figshare.22761149.v1. Each dataset consisted of snow, wet, and dry road-surface conditions; Table 1 provides more information on the two datasets.

The validation dataset was collected from other videos to test the feasibility of the proposed model. The original resolution of each image was $1{,}920 \times 1{,}080$ or $1{,}280 \times 720$. The color space of each image was red, green, and blue (RGB). Before the proposed model was applied, all the images were resized to $224 \times 224$, and each color space was considered. Additionally, histogram-based image equalization was applied to the images. For the YouTube-w-ALI dataset, 10,193 images were chosen as the training set, while the remaining 10,781 images were chosen for testing. For the YouTube-w/o-ALI dataset, 11,153 images were selected for training, while the remaining 13,073 images were selected for prediction.

Road-surface condition recognition is a type of material image recognition. Hence, to further verify the robustness of the MBFN model, we used the MattrSet dataset (*Zhang et al., 2021b*) to complete a four-fold cross-validation. This dataset was derived from real commodities online, including bags and shoes, and it was constructed under the guidance of experienced materials experts. It is a coarse-grained material dataset that includes material images of polyurethane (PU), canvas, nylon, and polyester. It contains 11,021 images with resolutions ranging from $123 \times 123$ to $4{,}300 \times 2{,}867$, making the material recognition task more challenging; Table 2 contains more information about this dataset.

In summary, the three datasets had apparent differences in the imaging mode, acquisition time, and sampling source. These results validated the effectiveness and

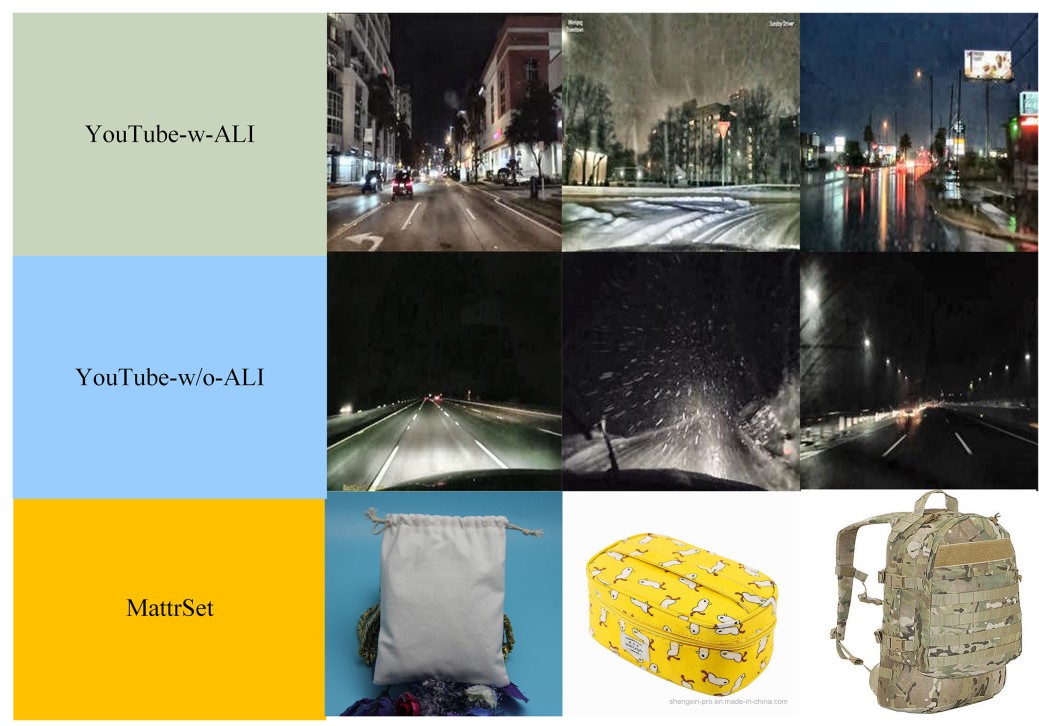

**Figure 5** **Representative images from each dataset.** Image source: Mattrset and YouTube (https://www.youtube.com/playlist?list=PLZi94yHOS-GROyCk6zvEnNb0LbCyCpc1O).

robustness of the proposed MBFN model. Figure 5 shows the representative images from each dataset.

# EXPERIMENTAL RESULTS AND DISCUSSION

## Baselines

We compare the proposed MBFN model with the following two types of baselines, most of them come from the literature (*Zhang et al., 2022*). Hence, this reference provides several key baselines. In our experiments, except the GPUs are different (our GPU is GTX2080Ti whereas the reference uses RTX3080), we obey the same data partition of this literature. Therefore, the whole comparisons are fair, which can better validate the effectiveness or superiority of our model.

1) Fine-tuned deep learning networks, which include CNNs with three, two, and one convolutional layers (*Zhang et al., 2022*), VGG-16 (*Simonyan & Zisserman, 2014*), VGG-19 (*Simonyan & Zisserman, 2014*), SqueezeNet (*Iandola et al., 2016*), ResNet50 (*He et al., 2015*), a Vision Transformer (*Dosovitskiy et al., 2020*), DenseNet121 (*Huang et al., 2016*), and ConvNeXt (*Liu et al., 2022*).

2) Two coarse-grained material image recognition models: saliency-excitation coarse-grained segmentation (SECGS) (*Zhang et al., 2021b*) and hierarchical materials factorization framework (HMF$^2$) (*Zhang et al., 2020*).

**Table 3 Comparisons with different baselines.**

| Models | YouTube-w-ALI | YouTube-w/o-ALI |
|---|---|---|
| Three convolution layer CNN | 90.08% | 90.96% |
| Two convolution layer CNN | 90.72% | 90.16% |
| One convolution layer CNN | 90.89% | 89.96% |
| SqueezeNet model | 89.14% | 93.59% |
| VGG16 | 90.65% | 91.65% |
| VGG19 | 90.17% | 91.79% |
| ResNet50 | 92.54% | 92.17% |
| DenseNet121 | 94.08% | 95.46% |
| Vision transformer | 96.08% | 96.34% |
| MBFN | 97.28% | 97.71% |

## Experimental settings

We used PyTorch on a server with four NVIDIA GeForce GTX2080Ti GPUs with 94 GB of RAM. Then, we used the Adam optimizer with a weight decay and set the initial learning rate to 0.0005. The momentum value was 0.9, the regularized weight decay was 1e−4, and the learning rate decay was 0.1. The batch size was set to 16, and the proposed model was trained for 50 epochs. An accuracy metric was used to evaluate each recognition model.

## Quantitative results

First, we compared the MBFN model with all the baseline models. The corresponding experimental results are presented in Table 3. The proposed MBFN model performed well on both the datasets. We conducted an in-depth analysis from the following perspectives.

The performance of the MBFN model on the two road-surface image datasets was higher than those of the fine-tuned deep learning networks. For example, compared to the performance of the DenseNet121 model (most competitive), the MBFN model exhibited a performance improvement of 3.20% and 2.25% on the YouTube-w-ALI and YouTube-w/o-ALI datasets, respectively. Overall, the MBFN model was more effective and robust for road-surface condition recognition.

Deep learning networks, such as DenseNet121 and VGG16, are usually complex and need sufficient high-quality training samples, which may be prone to overfitting. Particularly, in road-surface condition recognition, most images are of low quality because of various weather conditions, and contain various types of noise interferences (see Fig. 5). Hence, deep learning baseline models usually receive inadequate training, which affects their final accuracy. Second, owing to the down-sampling structures of these networks, the recognition model only focuses on the local area, ignoring important contextual semantic information.

The MBFN model handled this problem better, achieving a better performance on the YouTube-w/o-ALI dataset. It used the ConvNeXt module to extract basic features, ensuring a basic discriminative ability. Then, it extracted more discriminative information through the bidirectional fusion module, learning semantically complementary and robust

**Table 4 Efficiency and parameters of different ConvNeXt backbones.**

| Model | Accuracy on different datasets (%) | | Params (MB) |
|---|---|---|---|
| | YouTube-w/o-ALI | YouTube-w-ALI accuracy | |
| ConvNeXt-T | 96.23 | 96.08 | 29 |
| ConvNeXt-S | 97.71 | 97.28 | 50 |
| ConvNeXt-B | 97.82 | 97.35 | 89 |

features at different scales. Additionally, multi-supervision was used to enable the two modules to be internally self-constrained and thereby achieve the best results. These factors contributed to reducing the negative impact of image quality on recognition and improving the final performance.

In order to ensure the effectiveness and lower the size of our model, we choose ConvNeXt-S as the backbone network of the proposed approach. Thus, we can obtain a good trade-off between recognition performance and model size through this way. More importantly, if the proposed road surface condition recognition model is deployed on a resource-constrained or edge device, it will become more significant that both real-time prediction and actual recognition performance all achieve satisfactory results.

To support our choice, we made a detailed comparison between different ConvNeXt backbones in the following Table 4. Each model obeys same data partitioning and was evaluated on the same computer server. As shown in Table 4, the ConvNeXt-B model is nearly 0.11% better than ConvNeXt-S, whereas the number of parameters in ConvNeXt-B is nearly 1.78 times of that in ConvNeXt-S. Compared with ConvNeXt-B, ConvNeXt-S is a "cheaper" choice. Similarly, compared with ConvNeXt-T, ConvNeXt-S is more effective and the corresponding number of parameters is relatively acceptable. Hence, based on this analysis, we consider using ConvNeXt-S as the backbone of the proposed model in terms of recognition performance and model size. Moreover, for the remaining two models including ConvNeXt-L and ConvNeXt-XL, their model sizes are 198M and 350M, respectively, which means that the parameters of the two models are too large. It will be harder to obtain a good trade-off between recognition performance and model size. So, we did not consider them in our comparison.

In conclusion, the MBFN model outperformed all the baseline models and was effective, robust, and widely applicable. There are still more challenges associated with clearly capturing and recognizing road-surface conditions in nighttime images. In the future, we intend to incorporate well-known attention mechanisms to better capture the key features in such images.

## Cross-validation results

Cross-validation is a useful method for objectively evaluating a proposed model, with the most used cross-validation method being the $k$-fold method. In this study, the training set was randomly divided into $k$ parts, and one part was used as the validation set to evaluate the MBFN model while the remaining $k - 1$ parts were used as the training set. This

**Table 5 Four-fold cross-validation on the MattrSet.**

| Model | 1$^{st}$ fold ↑ | 2$^{nd}$ fold ↑ | 3$^{rd}$ fold ↑ | 4$^{th}$ fold ↑ | Avg ↑ | Improvement | Std |
|---|---|---|---|---|---|---|---|
| ConvNeXt | 73.70% | 73.70% | 73.50% | 74.50% | 73.85% | 1.23% | 0.384% |
| SECGS | 70.10% | 71.38% | 72.10% | 72.30% | 71.42% | 3.66% | 0.862% |
| HMF$^2$ | 71.30% | 71.90% | 72.50% | 73.10% | 72.20% | 2.88% | 0.671% |
| MBFN | 74.80% | 74.70% | 74.60% | 76.20% | 75.08% | – | 0.653% |

Note:
"Improvement" is a relative performance improvement of the MBFN model compared to the corresponding baseline. Avg means average value. Std means standard deviation.

**Table 6 Four-fold cross-validation on the YouTube-w-ALI and YouTube-w/o-ALI.**

| Dataset | 1$^{st}$ fold | 2$^{nd}$ fold | 3$^{rd}$ fold | 4$^{th}$ fold | Avg | Std |
|---|---|---|---|---|---|---|
| YouTube-w-ALI | 97.34% | 96.86% | 96.53% | 97.35% | 97.02% | 0.345% |
| YouTube-w/o-ALI | 97.22% | 98.12% | 97.32% | 96.94% | 97.40% | 0.438% |

procedure was repeated $k$ times, and an average value was obtained. In this section, we have presented the evaluation of the MBFN model using a four-fold ($k$ = 4) cross-validation on the MattrSet dataset. The experimental results are listed in Table 5. We also performed a four-fold ($k$ = 4) cross-validation on the two road-surface condition datasets; the results are shown in Table 6.

As shown in Table 5, according to the lower standard deviation, the corresponding performances of each fold of the MBFN model were relatively close, and the average value was satisfactory.

The MBFN model outperformed each baseline model. As shown in Table 6, the MBFN model also achieved a stable performance on the two road-surface condition datasets. Overall, these results further demonstrated the good generalization ability and stability of the proposed model.

## Real-time running curves

In this section, we present the real-time running curves of several models, including ConvNeXt, the modified MBFN model that uses the concatenation method to fuse the basic and refined features (MBFN (Con)), the modified MBFN model that uses the addition method to fuse the basic and refined features (MBFN (Add)), the modified MBFN model that uses only refined features (MBFN (Sing)), and the modified MBFN model that uses the basic and refined features for multi-supervision without any fusion operation (MBFN(w/o Con)). The MBFN (Con), MBFN (Add), MBFN (Sing), and MBFN (w/o Con) models were regarded as variants of the proposed model. The corresponding results of these models are shown in Fig. 6.

First, as the number of epochs increased, the corresponding training and testing accuracies increased. Noteworthily, the models with multi-supervision converged faster than those without multi-supervision did. The proposed multi-supervised method improved the convergence speed of the MBFN model.

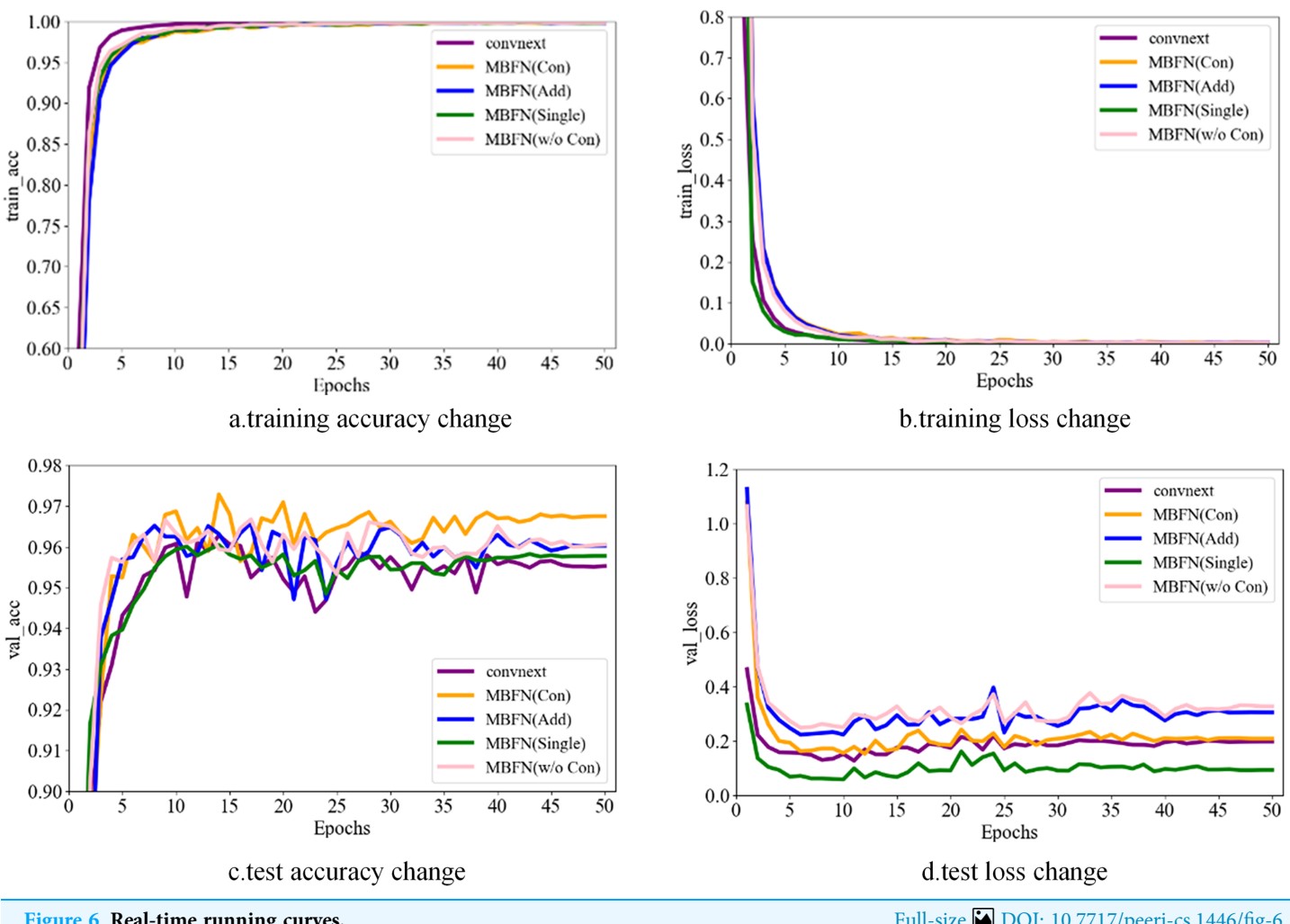

a.training accuracy change

b.training loss change

c.test accuracy change

d.test loss change

**Figure 6 Real-time running curves.**

Subsequently, the models were fully trained. The training loss of the MBFN model decreased rapidly and finally reached a stable value; please refer to MBFN (Con) in Fig. 6D. As shown in Fig. 6C, the accuracy curve of single supervision (MBFN (Sing) and ConvNeXt) was near the bottom after stabilization, whereas the curve of multiple supervision was generally higher, which was attributed to the role of multi-supervision in the model. Moreover, the concatenation method, which created refined features with a greater discriminative ability, was more valuable for improving the final accuracy. As shown in the figure, the effect of the model without the two-way fusion module was less than that of the model with the bidirectional fusion module. Moreover, the running curves demonstrated that the proposed model variants achieved a satisfactory recognition performance.

In summary, the MBFN model was fully trained and exhibited strong robustness and effectiveness in road-surface condition recognition.

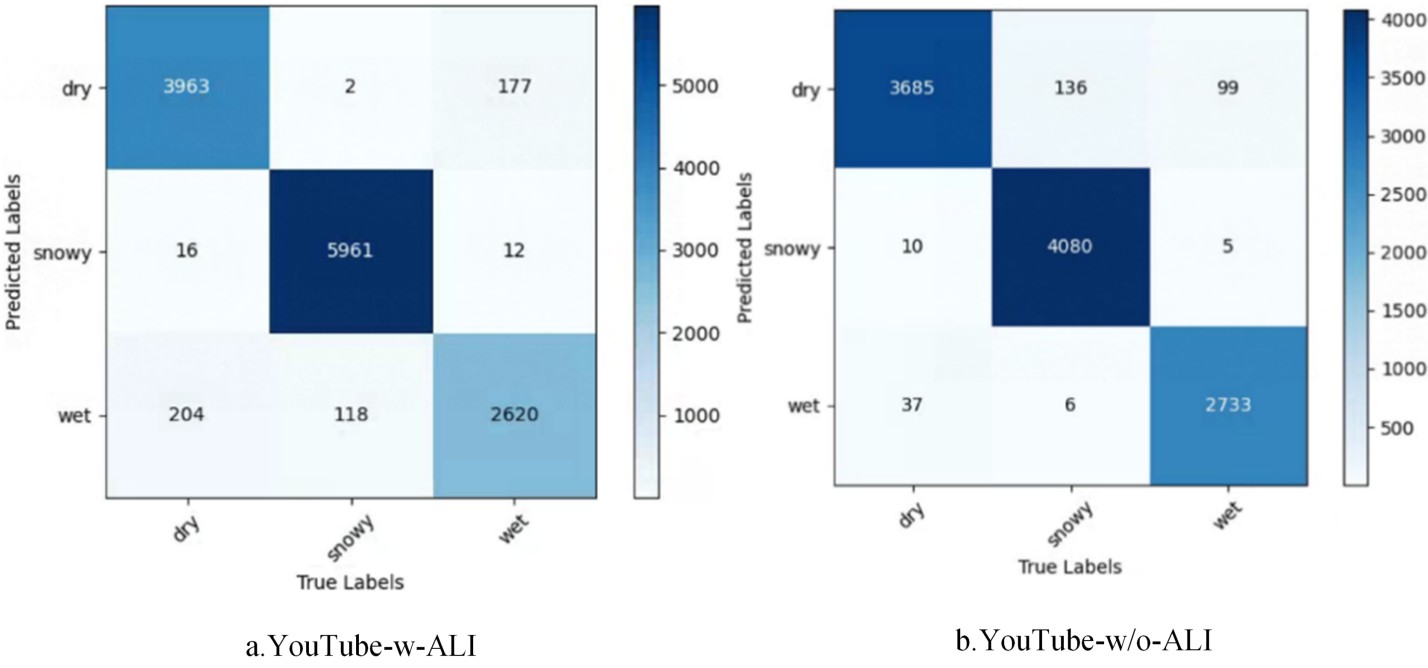

a.YouTube-w-ALI

b.YouTube-w/o-ALI

**Figure 7  Confusion matrix.**                                 

## Confusion matrix

In this section, we present the confusion matrices of our model, as shown in Fig. 7. On both the datasets, the proposed model recognized the snowy road-surface condition most accurately, followed by the dry and slippery road-surface conditions. The MBFN model could more accurately recognize the surface state of snowy roads, while it struggled in recognizing wet and slippery surface conditions since they produced reflection and scattering of light that increased the noise of the corresponding images. This phenomenon was more evident in the wet road-surface images. In our future work, more attention will be paid to wet and slippery road-surface conditions so that the MBFN model can provide a more satisfactory recognition performance for such conditions.

## Ablation analysis

The MBFN model consists of the ConvNeXt framework, a bidirectional fusion module, multi-supervised loss, and a feature fusion strategy. Each component plays a key role in improving the overall performance. Therefore, it was necessary to conduct detailed ablation analysis experiments to evaluate the actual contribution of each component. In this section, we present the corresponding ablation analysis results of these components on the YouTube-w-ALI and YouTube-w/o-ALI datasets. The results are shown in Table 7.

As shown in Table 7, the performance of each dataset was improved by the addition of a bidirectional fusion module. While the performance improved from 96.04% to 96.67% for YouTube-w-ALI, only a slight improvement was observed for YouTube-w/o-ALI. Furthermore, more performance improvements were observed when the multi-supervision method was used, particularly for (MBFN (Con)). The contribution of the

**Table 7 Ablation experimental results.**

| Case | Basic features | Bidirectional fusion | Multi-supervision (Add) | Multi-supervision (Concat) | YouTube-w-ALI | YouTube-w/o-ALI |
|---|---|---|---|---|---|---|
| A | ✓ | ✗ | ✗ | ✗ | 96.04% | 96.32% |
| B | ✗ | ✓ | ✗ | ✗ | 96.28% | 96.71% |
| C | ✓ | ✓ | ✗ | ✗ | 96.67% | 96.87% |
| Ours 1 | ✓ | ✓ | ✓ | ✗ | 96.98% | 97.71% |
| Ours 2 | ✓ | ✓ | ✗ | ✓ | 97.28% | 97.37% |

components can be ranked as follows: bidirectional fusion > multi-supervision > basic features. Moreover, the ablation results demonstrated that the proposed model variants achieved a satisfactory recognition performance.

The ablation experiments revealed that each component of the MBFN model was effective for road-surface condition recognition, with the MBFN model significantly outperforming the existing baseline models. In summary, the MBFN model effectively exploited its various components, becoming a model with a powerful generalization ability.

## CONCLUSIONS

In this study, we propose a novel model called MBFN for road-surface condition recognition. Our model comprises the ConvNeXt, bidirectional fusion, and multi-supervised loss modules. We validated the effectiveness and robustness of the MBFN model on two datasets including daytime and nighttime images. The proposed model displayed certain advantages in road-surface condition recognition. Moreover, the generalization ability of the MBFN model on two road-surface datasets and one coarse-grained material dataset was demonstrated through cross-validation experiments. The data from the results can improve the reliability of road-condition detection in both urban and suburban areas, such as rural areas or highways.

In the future, we will strive to improve the quality of nighttime images, potentially alleviating the problem of noise interference and building a firm foundation for road-surface condition recognition. We also plan to employ advanced transformer (*Vaswani et al., 2017*) to explore new research directions and enrich the original benchmark datasets by including other road-surface conditions.

## ACKNOWLEDGEMENTS

The authors would like to thank the editor and the reviewers for their helpful suggestions.

### Funding

This research was funded by the National Natural Science Foundation of China (Grant No. 62161011), the Key Research and Development Plan of Jiangxi Provincial Science and Technology Department (Key Project) (Grant No. 20223BBE51036), the Training Plan for Academic and Technical Leaders of Major Disciplines of Jiangxi Province (Grant No.

20204BCJL23035), the Science and Technology Projects of Jiangxi Provincial Department of Education (Grant No. GJJ200628). There was no additional external funding received for this study. The funders had no role in study design, data collection and analysis, decision to publish, or preparation of the manuscript.

## Grant Disclosures

The following grant information was disclosed by the authors:
National Natural Science Foundation of China: 62161011.
Key Research and Development Plan of Jiangxi Provincial Science and Technology Department: 20223BBE51036.
Training Plan for Academic and Technical Leaders of Major Disciplines of Jiangxi Province: 20204BCJL23035.
Science and Technology Projects of Jiangxi Provincial Department of Education: GJJ200628.

## Competing Interests

The authors declare that they have no competing interests.

## Author Contributions

- Hongbin Zhang conceived and designed the experiments, performed the experiments, analyzed the data, performed the computation work, prepared figures and/or tables, authored or reviewed drafts of the article, and approved the final draft.
- Zhijie Li conceived and designed the experiments, performed the experiments, analyzed the data, performed the computation work, prepared figures and/or tables, authored or reviewed drafts of the article, and approved the final draft.
- Wengang Wang conceived and designed the experiments, performed the experiments, analyzed the data, performed the computation work, prepared figures and/or tables, authored or reviewed drafts of the article, and approved the final draft.
- Lang Hu conceived and designed the experiments, authored or reviewed drafts of the article, and approved the final draft.
- Jiayue Xu conceived and designed the experiments, prepared figures and/or tables, and approved the final draft.
- Meng Yuan performed the experiments, authored or reviewed drafts of the article, and approved the final draft.
- Zelin Wang conceived and designed the experiments, prepared figures and/or tables, and approved the final draft.
- Yafeng Ren performed the experiments, performed the computation work, authored or reviewed drafts of the article, and approved the final draft.
- Yiyuan Ye analyzed the data, authored or reviewed drafts of the article, and approved the final draft.

## Data Availability

The code is available at Zenodo: NLPLitter. (2023). NLPLitter/MBFN: code_MBFN (v1.0.0). Zenodo. https://doi.org/10.5281/zenodo.7896316.

The videos are available at figshare: 李，志杰 (2023). veido. figshare. Media. https://doi.org/10.6084/m9.figshare.22775078.v1.

The resized and pre-processed dataset is available at figshare: 李，志杰 (2023). dataset. figshare. Dataset. https://doi.org/10.6084/m9.figshare.22761149.v1.

The MattrSet is available at figshare: 李，志杰 (2023). mattrset dataset. figshare. Dataset. https://doi.org/10.6084/m9.figshare.22774991.v1.

## Supplemental Information

Supplemental information for this article can be found online at http://dx.doi.org/10.7717/peerj-cs.1446#supplemental-information.

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
