# Peer review of "Multi-supervised bidirectional fusion network for road-surface condition recognition"

_PeerJ Computer Science, doi:10.7717/peerj-cs.1446_

## Round 0.1 · original submission · Major Revisions

While the reviewers found the paper interesting, they are all agreed that very extensive revisions will be necessary if the paper is to be acceptable, to make the paper of interest and benefit to a wider cross-section of our readers.

Please see below a copy of the suggestions of the reviewers, which need to be addressed before re-submission. I hope that you will find the reviewers remarks pertinent and helpful; we look forward to receiving a revised version of the paper.

Reviewer 1 ·

Basic reporting

In this paper, a novel model for road surface condition recognition is proposed. My concerns are as follows:
1. I would like to know why the author chose ConvNeXt S among five different scales of ConvNeXt models?
2. I suggest briefly explaining the role of FPN in section 3.2, which will be more friendly to scholars who have just entered this field

Experimental design

3. The author should introduce the key parameter settings of SqueezeNet and VIT.
4. If a portion of the experimental results comes from reference [29], the author should state this and state that the experimental conditions are consistent with it.

Validity of the findings

No

Additional comments

No

Reviewer 2 ·

Basic reporting

This paper provides a novel model named MBFN for road surface condition classification. There are many formatting errors in the article. There are also some modifications to be made in the content of the article.

1. Line 62, "(CV)" is not used in the rest of this article. Such abbreviations are unnecessary.
2. Cite references normatively and put them all at the end of the sentence. For example, line 104-105 "Manuel Amthor [37] et al. proposed a texture-based model to detect the wet road conditions." should revised as "Manuel Amthor et al. proposed a texture-based model to detect the wet road conditions [37]."
3. Figure 4 need to be revised. There is something wrong with the arrows in the figure.
4. The meaning of formula 3 does not agree with that described by the author. The author needs to go over it and fix it. Formula 4 is not centered, and the font size is not consistent with the previous three formulas.
5. There are so many errors in the format of the references that they need to be carefully revised. For example,
“[1] Zhang, H.; Azouigui, S.; Sehab, R.; Boukhnifer, M.; Balembois, F.; Bedu, F.; Cayol, O.; Beev, K.; Planche, G. Remote sensingtechniques to recognize road surface conditions for autonomous vehicles. In Proceedings of the SIA VISION, Paris, France, 17–18March 2021; pp. 179–184.”
“[2] Bellone, M.; Ismailogullari, A.; Müür, J.; Nissin, O.; Sell, R.; Soe, R.M. Autonomous Driving in the Real-World: The Weather Challenge in the Sohjoa Baltic Project. In Towards Connected and Autonomous Vehicle Highways; Springer: Berlin/Heidelberg,Germany, 2021; pp. 229–255.”
“[3] European Road Safety Observatory. Annual Accident Report 2018; European Road Safety Observatory: Brussels, Belgium, 2018.”

Experimental design

no

Validity of the findings

no

Additional comments

no

---

## Round 0.2 · accepted · Accept

I think the author's revisions have met the requirements of reviewers and have reached the level of publication in this journal.